# What Factors Shape the Flyability in Bats?—The Perspective from Bat’s Wing Development

**DOI:** 10.3390/biology14111524

**Published:** 2025-10-30

**Authors:** Minjie Zhang, Hui Wang, Zhongzheng Liu, Mingyue Bao, Xintong Li, Tianhui Wang, Ruixue Wang, Jiang Feng

**Affiliations:** 1College of Life Science, Jilin Agricultural University, Changchun 130118, China; zhangminjie@mails.jlau.edu.cn (M.Z.); liuzhongzheng@mails.jlau.edu.cn (Z.L.); baomingyue@mails.jlau.edu.cn (M.B.); lixintong@mails.jlau.edu.cn (X.L.); wangtianhui@mails.jlau.edu.cn (T.W.); wangruixue@mails.jlau.edu.cn (R.W.); 2Jilin Provincial International Cooperation Key Laboratory for Biological Control of Agricultural Pests, Changchun 130118, China; 3Jilin Provincial Key Laboratory of Animal Resource Conservation and Utilization, Northeast Normal University, Changchun 130117, China

**Keywords:** bats, echolocation, powered flight, wing membranes, wing development

## Abstract

Bats are the only mammals capable of true powered flight, a remarkable ability supported by their highly specialized wing structures and complex physiological regulation. Herein, flyability is defined as a comprehensive biological trait supporting bat-powered flight. This review synthesizes current advances in understanding the development of bat wings and flight capabilities, encompassing the evolution of wing morphology, flight-related muscle function, energy metabolism adaptations, and the developmental trajectory from embryonic to adult stages. We further discuss the co-evolutionary relationship between flight and echolocation, as well as the roles of genetic regulation, muscle dynamics, and environmental factors in shaping wing development. Existing research indicates that the molecular mechanisms underlying bat wing formation remain insufficiently understood, limiting our comprehension of the origins of mammalian flight and ecological adaptation strategies. This integrative analysis not only sheds light on the evolutionary mechanisms of flight but also provides theoretical foundations for species conservation and the design of bio-inspired aerial technologies.

## 1. Introduction

The origin of novel traits and their adaptive evolution has long been a central research focus in biological sciences, significantly enhancing our understanding of the mechanisms underlying species diversity [1,2]. To adapt to the vast nocturnal ecological niche, bats have evolved distinctive echolocation abilities and exceptional powered flight capabilities, making them one of the most evolutionarily successful animal groups [3,4]. Bats represent the second-largest order of mammals, surpassed only by rodents in terms of species diversity [5,6]. With their highly developed echolocation and sophisticated flight skills, bats are able to accurately capture prey in the night sky and play a crucial role in maintaining ecological balance through pollination [7,8,9], seed dispersal [10,11], and pest control [12,13,14].

Although bats have long been known for their exceptional echolocation abilities, not all bats possess this capability. For example, members of the suborder Yinpterochiroptera, such as Old World fruit bats (Pteropodidae), have highly developed vision and lack echolocation [15,16]. While a few taxa (notably *Rousettus*) exhibit a tongue-click, non-laryngeal modality that provides insights into incipient echolocation within the group [17,18]. Most bats rely on echolocation for navigation and hunting in low-visibility environments, enabling them to construct an acoustic image of their surroundings. As a result, they are widely regarded as a model species for research in the field of acoustics. The earliest research on bats using echolocation for environmental perception dates back to 1938, conducted by American zoologist Donald R. Griffin and his colleague Robert Galambos. In his study, Griffin first coined the term “echolocation” to describe the ability of bats to perceive their surroundings by emitting ultra-sonic waves and analyzing the returning echoes [19,20]. Based on differences in vocalization methods and frequencies, echolocating bats can be categorized into frequency modulation (frequency modulation, FM) bats, constant frequency–frequency modulation (CF-FM) bats, frequency-modulated, quasi-constant frequency (FM-QCF) pulse bats [21,22]. Researchers have not only provided a detailed classification of echolocation types but have also conducted extensive studies on the auditory structures, neurophysiology, molecular characteristics, and evolutionary mechanisms of different echolocating bat species [23,24,25,26,27]. These studies have significantly enriched our understanding of bat echolocation. However, compared to echolocation, another crucial phenotype in bats—flight—remains largely underexplored [28]. As evidenced by the disparity in the number of published studies, research on bat echolocation far surpasses that on powered flight and wing development (Figure 1). Echolocation, with its direct links to acoustic biology [29], neurophysiology [30], and behavioral ecology [31], has long been prioritized, likely due to its “novelty” as a mammalian sensory innovation and the relative ease of studying acoustic signals (e.g., via sound recording and playback experiments) compared to flight-related traits. Publications of bat echolocation may also drive by interests in sensory evolution and its applications in biomimetic acoustics, whereas flight research has remained fragmented, partly because flight capability is a complex phenotype integrating anatomy, biomechanics, physiology, and development, making it more challenging to study with a single experimental approach. The slower growth in flight-related publications also reflects a historical bias toward “observable” traits (e.g., echolocation calls) over traits requiring interdisciplinary methods (e.g., wing membrane biomechanics or embryonic development tracking).

Notably, Table 1 and Figure 2 further highlight the limitations of bat wing development research. The low number of publications (far fewer than both echolocation and general flight studies) reveals an additional layer of study bias, neglect of developmental mechanisms underlying key adaptive traits. Unlike echolocation research (where neurophysiological and behavioral studies can be conducted on adult individuals) or general flight research (where biomechanical analyses can focus on adult wing function), wing development research requires access to bat embryos or juveniles, samples that are difficult to obtain for many bat species. Their nocturnal lifestyles, cryptic roosting places, and other unanticipated sampling restrictions may lead to limited research attention on bat wing development.

This limited number of studies suggests that research on bat developmental biology remains insufficiently understood or inadequately addressed. We searched the Web of Science database using the keyword “bat flight” and summarized the accessible studies with respect to their study species, case examples, research objectives, methodologies, and conclusions, and thematic categories (e.g., aerodynamics and kinematics, energetics and metabolism), as presented in Appendix A. Noting that aerodynamics research has increasingly focused on unsteady aerodynamic mechanisms and species-specific kinematics in recent years, while energetics studies have centered on physiological adaptations and environmental modulation of flight energy costs. Meanwhile, morphology and development studies have spanned organismal and molecular scales: at the organismal level, they have tracked postnatal wing growth milestones and quantified how juvenile wing development shapes flight performance, while at the molecular level, they have uncovered gene expression shifts and epigenetic regulation that drive wing morphological specialization for flight. Ecology-related studies have emphasized the link between wing morphology (e.g., aspect ratio, wing loading) and niche partitioning, including vertical habitat stratification (e.g., clutter vs. open-space occupation) and dietary specialization, and further explored how flight traits support species coexistence and adaptation to diverse environments.

All extant bat species possess well-developed flight capabilities and have evolved wing structures adapted specifically for powered flight. Hence, the order Chiroptera is used to collectively refer to bats, highlighting this representative characteristic of the group [35,36]. Moreover, it appears that the ancestors of bats also had the ability to fly [37,38]. The bat fossils of *Icaronycteris*, discovered in Wyoming, USA, in 2003, are currently considered the earliest known bats. These fossils date back to the early Eocene, approximately 52.5 million years ago [39]. The earliest fossil record of a mammal with wing membranes was reported by Meng et al. in 2006, following the discovery of a mammalian fossil with wing membranes in Inner Mongolia, China [40]. Analysis revealed that this mammal, which lived approximately 125 million years ago, already possessed the ability to glide. This mammal had specialized dentition and moderately sized wing membranes. The wing membranes were covered with fur and supported by the limbs and tail. As the earliest known gliding mammal in the world, its discovery has pushed back the history of flying mammals by nearly 80 million years. Since the discovery of bats, powered flight has been considered one of their most distinctive characteristics. Simons et al. [39] proposed that bats first evolved the ability to fly, followed by the evolution of echolocation. This hypothesis is based on the anatomical features of the ear region in the fossil specimen of *Onychonycteris finneyi*, which do not support echolocation capabilities, while the overall body structure is already adapted for powered flight. It is worth noting that Onychonycteris exhibits limb proportions consistent with a scansorial bauplan rather than the typical chiropteran condition; while the metacarpals and phalanges are appreciably elongated, they are still shorter than in Icaronycteris. Furthermore, it must be acknowledged that its flight was likely limited to short inter-tree movements, with constrained speed and maneuverability. However, interpretations of *O*. *finneyi*’s functional capabilities remain debated, with conflicting evidence regarding whether it possessed echolocation, highlighting key challenges in inferring soft-tissue-dependent traits from fossil morphology [41,42,43]. This debate underscores a broader methodological limitation in fossil bat research: soft-tissue structures central to echolocation, such as the larynx, cochlear hair cells, and auditory nerves, are extremely rarely preserved in vertebrate fossils. Without direct evidence of these tissues, inferences about echolocation rely on indirect bony correlates, which can be ambiguous. Such uncertainty does not negate the fossil’s value, Giannini et al.’s (2024) aerodynamic model clearly demonstrates its powered flight capability [43], but it highlights the need to frame fossil interpretations as hypotheses rather than definitive conclusions. This ongoing controversy also reflects differing priorities in evolutionary inference.

The powered flight of bats is a complex aerodynamic process [44], primarily achieved by altering the shape of the wings during the upstroke to reduce the surface area of the wing membrane, bringing the wings closer to the body compared to the downstroke [45]. The flight capability of Chiroptera appears to be innate; according to Gregg F. Gunnell and Nancy B. Simmons in 2005 [46], no intermediate fossil forms have been found to link bats to gliding or non-flying mammalian ancestors. Giannini et al. reconstructed an aerodynamic flight model of the ancient [43], complete bat fossil *Onychonycteris finneyi*, revealing that bats were capable of both gliding and powered flight, whether in standard modern air density or the higher-density atmosphere of the Eocene. This suggests that this lineage of mammals with unique body structures may have appeared suddenly [46,47]. The lack of robust transitional fossil evidence makes determining the evolutionary position of bats based on morphological traits challenging and controversial in mammalian evolutionary studies [48].

This review systematically summarized the characteristics of powered flight in bats from the aspects of specialized wing structure, physiological adaptations for flight, and molecular regulation of wing development. Additionally, we also highlight the existing gaps in current research, particularly the limited understanding of the gene regulatory mechanisms involved in the development of bat wing membranes. Based on these gaps, the review suggests that future research should focus more on the co-evolutionary mechanisms between flight and echolocation, especially in areas such as gene network regulation, neural control of flight muscles, and the effects of environmental factors on development. These insights are crucial for understanding the ecological adaptation strategies and evolutionary mechanisms of bats.

## 2. Fundamental Requirements for Powered Flight

The realization of bat flyability relies on multiple fundamental biological adaptations. From the structural basis of flight, including wing membranes and skeletal modifications, to the physiological support, including muscle function and energy metabolism, these factors collectively determine the aerodynamic efficiency and flight performance of bats. To date, research has identified at least seven extant mammalian species that have independently evolved the ability to fly or glide [49]. This capability is attributed to the convergent evolution of wing membrane structures across distinct mammalian lineages [49]. Common examples of mammals capable of aerial flight or gliding include bats (Chiroptera) [43,49,50,51], the sugar glider (*Petaurus breviceps*) [49], the feathertail glider (*Acrobates pygmaeus*) [49,52], the Philippine colugo (*Cynocephalus volans*) [53], the Sunda colugo (*Cynocephalus variegatus*) [54], Siberian flying squirrel (*Pteromys volans*) [55], the northern flying squirrel (*Glaucomys sabrinus*) [56], etc. These mammals share a distinctive feature—a membrane of skin which stretches between the forelimbs and hindlimbs [49,57,58], resembling a gliding wing when extended (Figure 3). Thus, the plagiopatagium serves as the structural foundation for flight and gliding behaviors in mammals [59,60,61], making the study of the genes involved in the development of this membrane an intriguing topic in current research.

This raises a key scientific question: what are the developmental and evolutionary mechanisms underlying the formation of the patagium in gliding or flying mammals? Moreno et al. [65] identified *Emx2* as a key molecular driver of patagium development in gliding marsupials, acting through independently evolved, convergent regulatory elements across distinct lineages. This study focuses on elucidating the developmental and evolutionary mechanisms underlying a phenotypic innovation—the marsupial gliding membrane (patagium)—with particular emphasis on the convergent molecular basis by which this structure has independently arisen multiple times, including the identification of core regulatory genes and their upstream cis-regulatory elements. The study reveals that the key transcription factor *Emx2* is upregulated in all three independently evolved gliding marsupials, with lineage-specific genomic accelerated regions (GARs) identified in its surrounding cis-regulatory landscape that significantly enhance *Emx2* expression. In vivo shRNA-mediated knockdown experiments demonstrated that *Emx2* is essential for patagium formation, acting through direct regulation of downstream signaling pathways—most notably *Wnt5a*—to promote mesenchymal cell proliferation and epidermal thickening, both critical cellular processes in gliding membrane development. This study suggests that natural selection may drive morphological convergence in mammals by recruiting non-homologous enhancers across distinct lineages to regulate a shared developmental regulator.

In a comparative study of limb development between bats and mice, researchers have delved into the cellular origins and molecular mechanisms underlying the formation of the bat patagium, with the goal of pinpointing the developmental stage at which bat wing development begins to diverge from the canonical mammalian limb pattern. Schindler et al. [66] have made significant contributions to this field by elucidating the cellular and molecular underpinnings of the bat forelimb membrane, known as the chiropatagium. Their work has revealed that the wing membrane is predominantly composed of a fibroblast population characterized by high levels of *MEIS2* expression. Building on this foundation, it has been demonstrated that *MEIS2* extensively binds to distal forelimb regulatory domains and directly activates a fibroblast program that includes *TBX3* and *TBX18*. This finding highlights a bat-specific regulatory architecture that exhibits minimal overlap with the promoter regions of early mouse limbs. Functionally, the distalization of *MEIS2* and *TBX3* in transgenic mice, driven by a *Bmp2* distal limb enhancer, induces wing-related phenotypes in the distal mesenchyme and interdigital domains [66]. This observation supports the hypothesis that a proximal developmental program has been repurposed for distal development during the evolution of the chiropatagium. Importantly, this fibroblast population does not originate from the canonical programmed cell death region (RA-Id), but rather follows a distinct developmental trajectory. The transcription factors *MEIS2* and *TBX3* have been shown to govern a novel gene expression program in these cells, which closely resembles the transcriptional network utilized by proximal limb fibroblasts in mice. Collectively, these findings suggest that bats have evolutionarily redeployed a conserved proximal limb gene program to a distal domain, thereby facilitating the emergence of the chiropatagium as a morphological innovation.

### 2.1. Unique Powered Flight in Bats

Among the mammalian groups capable of flight or gliding, the flight behavior of bats is particularly distinctive. It is referred to as powered flight because bat flight represents true flight in the strictest sense [48,49]. Studies [67] utilizing computational fluid dynamics (CFD) and high-resolution motion capture technology have demonstrated that bats can generate complex wing vortex structures during flight. These structures help bats enhance flight efficiency and avoid net thrust loss between the upstroke and downstroke, indicating that while bats may not achieve the same flight speeds as birds [68,69,70], they exhibit remarkable agility and control over flight speed. The wing structure of bats is composed of flexible and elastic bones, along with multiple independently controllable joints, which allow them to generate complex dynamic wing morphologies. By regulating the movement of these joints and altering the deformation of the wing membrane, bats can adjust the shape of their wings during flight to optimize airflow utilization, thereby maintaining flight stability and maneuverability [71]. In contrast, mammals such as sugar gliders and flying squirrels can only achieve aerial movement through gliding by extending their patagium [72,73,74,75]. Taking the sugar glider as an example, its gliding usually begins with a leap from a high point, after which it quickly extends its wing membrane [76,77,78]. This wing membrane is composed of skin folds between the forelimbs, hindlimbs, and tail. By stretching these skin folds, they increase air resistance and maintain a relatively fixed posture during gliding, without significant limb movements [79]. The primary force for gliding comes from gravity rather than muscular activity. Their gliding path typically follows a smooth arc; upon leaping, they extend their wing membrane, using air resistance to slow their descent and control their direction. During gliding, the sugar glider’s body remains mostly stationary, relying on the extension of the patagium and gravity for navigation [49]. In contrast, bats possess significantly elongated forelimbs and a larger wing membrane area, with a key structural distinction from gliders lying in the attachment site of the proximal patagium to the forelimb, a detail critical to their flight capabilities. In gliding mammals (e.g., *P. breviceps* [49]), the proximal patagium attaches to the carpus; this arrangement preserves full mobility of the hand, allowing it to function independently for substrate support and climbing along tree trunks or branches [49,58]. In bats, by contrast, the proximal patagium does not attach to the carpus but instead integrates with the forelimb such that its interdigital segment, termed the dactylopatagium, stretched between digits 2–5 and forms the core propelling part of the wing [44,62,80]. This dactylopatagium is essential for powered flight: during the downstroke, its tight attachment to elongated digits 2–5 enables it to stretch and generate sufficient lift, while its elasticity allows for controlled deformation during the upstroke to reduce drag [45,71], a dynamic function absent in gliders, where the carpus-attached patagium acts only as a passive surface for gliding [58].

### 2.2. Specialized Flight-Related Muscles in Bats

For flight, birds require powerful and well-developed flight muscles attached to the keel for energy provision [81]. Similarly, bats have evolved unique flight-related muscles to enable powered flight, which may regulate membrane stiffness or serve other biomechanical functions [47,82]. Previous research by Foehring & Hermanson [83] discovered that in the Brazilian free-tailed bat (*Tadarida brasiliensis*), the subscapularis muscle primarily originates from the anterior surface of the scapula, attaches to the lesser tubercle of the humerus, and is responsible for medially rotating the upper arm during flight, aiding in wing retraction. Hermanson & Altenbach [84] further complemented this understanding through studies on the shoulder and arm musculature of the Jamaican fruit bat (*Artibeus jamaicensis*), elucidating the muscles involved in bat flight. The pectoralis major, originating from the sternum (specifically the manubrium) and inserting onto the crest of the greater tubercle of the humerus, serves as the primary downstroke muscle, contracting during the downstroke phase to drive the wings downward and provide thrust. The serratus ventralis thoracis, originating from the sternum and upper ribs and attaching to the medial edge of the scapula, assists in moving the scapula forward and stabilizing the wings during the downstroke. The latissimus dorsi, originating from the thoracic and lumbar vertebrae and the ilium and inserting into the intertubercular groove of the humerus, is an essential upstroke muscle that helps elevate the wings during the upstroke phase. The short head of the biceps brachii, originating from the coracoid process and supraglenoid tubercle of the scapula and attaching to the radial tuberosity, is involved in wing adduction during the downstroke phase. Additionally, they identified the trapezius group, which arises from the posterior skull, cervical, and thoracic vertebrae and attaches to the scapula and clavicle, aiding in stabilizing the shoulder girdle to support both upstroke and downstroke movements. Prior research [85] has demonstrated that the efficiency and complexity of bat flight largely depend on the fine division of labor and cooperation among these muscle groups. The unique wing muscles are crucial components for achieving powered flight in bats [47].

In addition, bats possess a distinct muscular complex known as the occipito-pollicalis (OP) muscle, which extends along the propatagium and connects to the anterior side of the carpus along the leading edge of the inner wing [47]. The OP muscle is essential for powered flight [47,86] and has both proximal and distal spatially separated precursors. The former is first detected as a small muscle mass located dorsally to the auricular muscles and sternocleidomastoid in stage 15 bat embryos. Tokita et al. [47], through detailed descriptions of the ontogeny of bat wing muscles such as the occipito-pollicalis and plagiopatagiales in 2012, revealed that bat wing muscles are derived from diverse embryonic origins, with a spatiotemporal correlation between wing membrane growth and muscle expansion (Figure 3). Furthermore, comparative gene expression analysis identified that Fgf10 signaling is uniquely activated in the primordia and mesenchymal tissues of the wing membrane, influencing the structural patterning of bat wing muscles [87].

Studies have further suggested that the specialized muscles at the wing tips of bats receive dual innervation from the facial nerve and the cervical spinal nerves [88]. Temperature has a significant impact on the performance of bat wing muscles [89,90]. Consequently, bat wing muscle tissues are relatively insulated, allowing higher body temperatures to maintain normal muscle contraction [91].

Tokita et al. [47] conducted a study on the development of bat wing muscles, providing a detailed anatomical analysis of the wing musculature. The study highlights the occipito-pollicalis muscle, which extends into the anterior wing membrane (propatagium). Additionally, the bat wing contains a series of muscles, including the coraco-cutaneus, humeropatagialis, and plagiopatagialis muscles located within the lateral wing membrane (plagiopatagium), as well as the uropatagialis muscle within the interfemoral wing membrane (uropatagium), all of which are essential for achieving powered flight.

### 2.3. Efficient Energy Supply and Antioxidant Defense

Flight is a high-energy-consuming process, and studies have shown that the energy expenditure of bats during flight is 3–5 times that of terrestrial mammals of similar body size [92,93]. Even the lowest metabolic demands of bats are significantly higher than the maximum metabolic capacities of terrestrial mammals [94]. Changes in energy mechanisms are likely the primary factor driving the evolution of flight within bat lineages [95,96,97]. Previous studies [92,98] measured oxygen consumption in bats during flight, further confirming that bats exhibit high metabolic rates while flying: the energy cost of bat flight is significantly higher than the cost of running for terrestrial mammals and is roughly similar to that of bird flight. Research by Guglielmo on bat fat metabolism [99,100] revealed that fat is the primary energy source during bat flight, with a significantly increased rate of fat breakdown during flight. Like birds, bats rely on fat as an energy source during flight [100,101,102]. The reliance on fat metabolism for sustained flight is a convergent trait between bats and birds, in that both taxa utilize fat’s high energy density to support prolonged aerial locomotion [99,103]. Fat provides a high amount of energy per unit mass and supports the metabolic demands of flight through the oxidation of fatty acids, making it an efficient fuel for such high-energy activities [99,100,104,105,106]. However, in most bats, fat reserves are relatively low compared to their needs, especially considering the high cost of flight [107,108,109]. Bats generally rely on strategies of rapid feeding and metabolism to meet their energy needs rather than storing large amounts of fat [110,111]. This approach minimizes the burden of excess weight, which would otherwise increase the cost of flight [112,113]. The elastic fibers in the wing membrane act like springs, capable of storing and releasing energy during wingbeats [114,115,116], thereby improving flight efficiency. This elasticity not only helps reduce energy consumption but also optimizes aerodynamic performance by adjusting the shape and stiffness of the wing membrane. The core “engine” of bat flight function lies in the adaptive changes in their unique mitochondrial energy supply system [95,117]. The interactions between mitochondrial and nuclear genes, along with compensatory evolution, may play a critical role in adapting to activities with high energy demands [118]. Studies have shown that during the evolution of flight capabilities in *Pteropus alecto* and *Eonycteris spelaea*, the mitochondrial oxidative phosphorylation genes underwent positive selection, with the mitochondrial respiratory chain providing 95% of the adenosine triphosphate (ATP) required for locomotion [119]. In bats, 23.08% of mitochondrially encoded oxidative phosphorylation genes and 4.90% of nuclear-encoded oxidative phosphorylation genes were found to be under positive selection, whereas only 2.25% of non-respiratory chain nuclear genes functioning in mitochondria or 1.005% of some other nuclear genes were under positive selection. Genes encoded by the mitochondria and the nucleus have undergone co-evolution to meet the increased energy demands associated with the origins of flight [95].

Notably, dietary guilds are a key yet underemphasized driver of bat flight energy metabolism, creating distinct adaptive strategies that align with ecological niches. Insectivorous bats rely primarily on fat oxidation to fuel flight, fat stores offer high energy density and sustained ATP production, critical for capturing agile prey in cluttered habitats or supporting long-distance migration. This strategy is further reinforced by their insect-based diet, which is rich in lipids and facilitates pre-flight fat accumulation [100,109]. In contrast, nectarivorous and frugivorous bats (e.g., *Pteropus poliocephalus* [34], *Cynopterus sphinx* [120]) have evolved to fuel flight directly with exogenous sugars, a trait linked to their high-sugar, low-fat diets [111]. These divergent strategies directly shape “flyability” that insectivorous bats’ fat dependency supports prolonged flight (e.g., 5–8 h of migratory flight [109]), while nectarivorous/frugivorous bats’ sugar-driven metabolism optimizes agility for frequent short flights (e.g., 10–15 min foraging bouts between fruit trees [120]). This contrast underscores that bat flight capability is not solely shaped by wing morphology, but also by dietary adaptations that match ecological demands.

Climate effects further modulate the energetic strategies underlying bat flight. In temperate regions, bats face seasonal temperature fluctuations that directly influence their flight metabolism: during cold springs, juvenile bats exhibit delayed postnatal growth of flight muscles and upregulated expression of heat shock proteins (HSP70/90), which help maintain muscle function at lower body temperatures [119]. This physiological adjustment, driven by climate constraints, extends the time required for juveniles to achieve sustained flight compared to tropical species [121,122]. For migratory bats, seasonal climate shifts also drive adaptive changes in fat storage. Prior to migration, these bats increase fat reserves, a response to the high energy demands of long-distance flight and the need to cope with unpredictable food availability along migration routes [109]. Such climate-mediated energetic adaptations highlight the interplay between physiological traits (energy metabolism) and ecological pressures (seasonal changes), a dimension critical to understanding the full scope of bat flight evolution. However, current conclusions on bats’ climate-driven energetic adaptations are largely limited to insectivorous and migratory insectivorous bats, with scarce data on frugivorous/nectarivorous bats, which may restrict generalizations about bat flight energy metabolism. Methodologically, the link between fat reserves and flight performance is often inferred indirectly (e.g., via body mass or tissue sampling) [123,124]. Therefore, acknowledging these limitations is important in framing energy metabolism conclusions to understand factors shaping bat flight.

The high energy and oxygen demand of flight may lead to an increase in reactive oxygen species (ROS) within bats, and purine metabolism serves as a crucial mechanism for coping with oxidative stress [125]. An imbalance between purine salvage and de novo synthesis pathways can produce harmful ROS. Under high oxidative stress conditions, enhanced purine metabolism involves molecular mechanisms such as positively selected genes, convergent changes, and non-parallel amino acid substitutions. Purine metabolism genes, including *ADA*, *AMPD3*, and *NT5E*, which exhibit significant adaptive changes in bats, have been identified. The expression of these genes in bats is significantly higher than in other mammals, highlighting their importance in antioxidative stress responses.

The high metabolic rates during flight often lead to an increase in bat body temperature, typically around 40 °C [126,127], which is much higher than that of non-flying mammals. To adapt to high body temperatures, bats exhibit elevated basal expression levels of heat shock proteins (*HSP70* and *HSP90*) across various tissues and cell lines. This characteristic has been validated in the Australian black flying fox (*P. alecto*) and the cave nectar bat (*E. spelaea*) [119]. Heat shock proteins (HSPs) are highly conserved master regulators of cellular stress, expressed in bat tissues and various cell lines. High levels of HSP expression enable bat cells to survive prolonged heat stress and other stressful conditions under which cells of other mammals would perish. This suggests that the expression of HSPs in bats may be a key factor in their adaptation to the high temperatures and metabolic pressures experienced during flight, potentially contributing to their longevity and disease resistance.

### 2.4. Specialized Wings Adapted for Powered Flight

First of all, the skeletal structure of bats shows significant adaptive modifications in the forelimbs [128,129,130]. The forelimb bones are highly specialized, characterized by the elongation of the second to fifth digits [71,130,131], reduced mineralization of the wing bones along the proximal–distal axis, and the presence of a novel muscle complex that controls and expands the interdigital membranes [132,133]. These specialized structures provide the necessary anatomical foundation for optimizing flight capabilities [34]. Crucially, bats have evolved specialized wing structures and physiological functions to facilitate powered flight. In 1960, R. C. Murphy [134] revealed the unique fluorescence properties of bat wing tissues by spreading bat wings coated with fluorescent dye and analyzing the wing structure under different wavelengths of light. In 1970, Cortese et al. [135] used high-intensity zirconium light to illuminate flattened bat wings and, under a microscope with 2400× optical magnification, observed the wing’s sebaceous and apocrine glands and associated structures. They discovered that bat hair, hair follicles, and sebaceous glands are encased in a specialized sheath. The secretion activity of the sebaceous glands is regulated by contractile fibers within this sheath, which rhythmically contract at about four times per minute to promote the secretion of sebum. Apocrine glands expel their secretions onto the skin surface through the contraction of the myoepithelial cell layer. The sheath surrounding hair follicles and sebaceous glands not only provides protection but may also be involved in the secretion process of sebum, although the exact function remains to be fully understood.

The patagium of bats is structurally complex and functionally unique, serving as a critical component of their powered flight capabilities [136,137,138]. The patagium is composed of skin and elastic fibers, connecting the forelimbs, hindlimbs, and tail, thereby forming a large surface area crucial for flight [86,137]. This structure is exceptionally thin and flexible, with a thickness-to-chord ratio of approximately 0.2% and a Young’s modulus of around 1 MPa [62,68]. These characteristics impart significant elasticity and lightweight properties to the patagium [44,139,140], enhancing its mechanical performance by allowing substantial deformation during flight to optimize aerodynamic efficiency [140]. The coordinated up-and-down flapping of the wings generates lift, while the flexible adjustment of the patagium’s shape facilitates fore-and-aft movements (cranial and caudal) [141,142], greatly enhancing flight agility and efficiency.

The patagium is divided into different regions, including the plagiopatagium, propatagium, dactylopatagium, and uropatagium [50,80]. The extensive expansion of the plagiopatagium contributes to the curvature development near the wing’s center of lift [62]. Besides the plagiopatagium, the well-developed interdigital membranes and the uropatagium [47,50] play essential roles in maintaining flight stability and directional control. The enhanced load-bearing capacity of the uropatagium is closely associated with its critical role in capturing insect prey. The bat’s patagium is closely integrated with the body, extending from the neck to the ankle, forming a “wing-body integration” structure [143]. During flight, bats can adjust the effective angle of attack of the wings by moving their legs up and down, even though this integrated structure somewhat restricts the wings’ range of motion [144,145,146,147,148]. Across chiropteran lineages, the relative area, chordwise tension, and stiffness of these elements vary, producing taxon-specific differences in leading-edge vortex stability, stall margin, and maneuvering control during flight.

Notably, the biomechanical properties of bat wings, including membrane flexibility and reduced bone mineralization, are not uniform across species but are tightly linked to habitat constraints [34,62]. Bat species inhabiting cluttered environments (e.g., tropical rainforests) exhibit relatively thinner and more elastic plagiopatagia, which enhance wing maneuverability for navigating between dense vegetation [149]. In contrast, open-space foragers have stiffer wing membranes and longer aspect ratios, optimizing gliding efficiency during long-distance flight [83]. These habitat-driven biomechanical differences directly reflect ecological adaptation: cluttered habitats select for high maneuverability to avoid obstacles and capture prey in confined spaces, while open habitats favor energy-efficient flight to cover large foraging ranges. Such ecological constraints further shape the developmental trajectory of wing structures. For instance, forest-dwelling bats show accelerated postnatal growth of the dactylopatagium (interdigital membrane), a trait that reinforces their biomechanical advantage in cluttered environments.

## 3. Wing Development: A Crucial Evolutionary Adaptation for Powered Flight

The foundation and key to understanding flyability in bats lie in the study of their wings, as the complex and unique wing-hand structure of bats provides them with exceptional powered flight capabilities [44,150]. From embryonic flank-limb fusion to postnatal wing growth, each stage of wing development lays the anatomical foundation for the acquisition and optimization of flight capabilities. Therefore, research on the growth and development of bat wing-hands is of particular importance. Studies have shown that the development of bat wing-hands begins during the embryonic stage and continues throughout the entire postnatal ontogenetic process [133,151], regulating by multiple genes and associated signaling pathways [33,152,153].

### 3.1. Wings Development During the Embryonic Stage Prepares for Powered Flight

The unique wing membrane structures of bats, such as the dactylopatagium and plagiopatagium, are preserved throughout embryonic development, playing a crucial role in enabling powered flight. A study by Anthwal et al. [50] provided significant insights into the formation of the bat plagiopatagium, revealing that it develops through a conserved process of embryonic flank-limb fusion. Post-fusion differential growth and gene expression drive ecological adaptation and morphological diversification. The initial formation of the wing membrane exhibits a highly consistent temporal pattern across different bat species. This conserved developmental program, involving flank-limb fusion, is primarily driven by localized inhibition of apoptosis in the early limb skin, rather than arising as a novel derivative of the skin. In bats, this fusion process is tightly regulated by the control of periderm development within the epidermal epithelium. Specifically, researchers observed no signs of the plagiopatagium at embryonic stage 14; however, by stage 15, the prospective plagiopatagium begins to grow from the lateral surface of the trunk. By stage 16, it has successfully fused with both the forelimbs and hindlimbs.

Previous studies on the short-tailed fruit bat *Carollia perspicillata* have described the stages of embryonic development, providing a detailed framework for the species’ embryonic staging, which serves as a reference for subsequent research [154]. With the advent and application of high-throughput sequencing technologies, researchers have identified key genes involved in regulating wing membrane development in bats through RNA-Seq analysis. These genes primarily govern the development of epidermal epithelial cells, including Ripk4 and Klf4. For instance, Ripk4 plays a crucial role in keratinocyte differentiation [155] and is essential for proper mammalian development, as mutations in this gene in humans are associated with the autosomal recessive Bartsocas-Papas syndrome [156]. Studies on 16 bat species, along with focused investigations on the insectivorous bat *Pteronotus quadridens* and the omnivorous bat *Erophylla sezekorni*, indicate that these genes begin to express at embryonic stage 15, initiating the early growth and subsequent fusion of the plagiopatagium. Furthermore, a study [136] utilizing whole-genome mRNA sequencing and in situ hybridization on embryonic limb tissues of bats and mice identified seven key genes exhibiting unique expression patterns in bat wings and feet. These genes include the 5′ *HoxD* cluster (*Hoxd9–13*), *Tbx3*, and *Fam5c*. The study found that the *Hoxd9–13* genes display significantly elevated and prolonged expression in the elongation regions of bat wings. Similarly, the Tbx3 gene is highly expressed over an extended period in the same regions, contributing to the inhibition of apoptosis and promoting cell proliferation, thereby facilitating wing membrane growth. The development of interdigital membranes in bats becomes evident at embryonic stage 14, during which the *Hoxd9–13* genes exhibit high expression in the rapidly elongating tissues of the digits and interdigital regions. This elevated expression persists through late stage 19, underscoring the critical role of these genes in the formation of bat interdigital membranes. Recent single-cell transcriptomic profiling of bat limbs reveals a forelimb-specific *PDGFD^+^* mesenchymal progenitor population that likely contributes to the interdigital membrane and stimulates bone-lineage proliferation [157]. Importantly, prolonged chondrogenesis with delayed osteogenesis increases chondrocyte proportions while reducing osteoblasts in developing forelimbs, coordinated by Notch activation together with suppression of WNT/β-catenin signaling.

Additionally, the well-known morphological factor *Wnt5a* has been identified as being involved in the growth and development of lateral body membranes in flight- or gliding-capable mammals, playing diverse roles throughout evolutionary developmental processes [49]. *Wnt5a* regulates multiple signaling pathways, controlling a broad range of cellular processes, including cell proliferation, differentiation, and apoptosis [158]. During embryogenesis, *Wnt5a* modulates the development of various structures as a secreted Wnt ligand, with its different isoforms exhibiting both anti-inflammatory and pro-inflammatory functions in cellular pathways and inflammatory diseases. It also plays a key role in driving distal-specific gene expression [159,160]. Essential for mammalian embryonic development, the absence of *Wnt5a* results in growth defects across multiple structures, including the intestine [161]. A 2023 study by Feigin et al. [49] on the lateral membranes of sugar gliders and bats identified a shared ancestral gene, *Wnt5a*, responsible for regulating the growth, development, and apoptosis of these membranes [49,158]. Feigin et al. further demonstrated that most extant mammals have inherited a developmental program driven by *Wnt5a*, which guides wing membrane formation through the proliferation and differentiation of mesenchymal cells. Specifically, *Wnt5a* facilitates the condensation of dermal mesenchyme and the thickening of the overlying epidermis. Transgenic experiments with mice overexpressing *Wnt5a* exhibited early wing-like structures, with dermal condensation and epidermal hyperplasia. The phenotypes observed in early membrane differentiation in these transgenic mice reflect similarities with the ancestral mammalian lineage shared by sugar gliders, which emerged approximately 160 million years ago. These findings suggest that *Wnt5a* directs wing membrane development and that most modern mammals have inherited this *Wnt5a*-driven program. However, there remains a critical limitation in validating Wnt5a’s function specifically in bats. Although gene-editing experiments have established a causal relationship between *Wnt5a* and the differentiation of wing skin, scientists have yet to fully understand how dense, thickened skin regions transition into the thin, wide membranes required for flight. Undoubtedly, additional genes with currently unknown functions are involved in this process. Beyond the canonical FGF–BMP balance, the same study indicates that PTN and PDGF pathways are pivotal drivers of forelimb cell proliferation during wing morphogenesis, consistent with the identification of *PDGFD*^+^ progenitors in the forelimb [157].

### 3.2. Postnatal Wing Development Is Essential for the Full Realization of Powered Flight

While embryonic bats possess the primordial wing structures necessary for flight, the postnatal period is critical for the further maturation of these structures [121,122,162]. Previous research on the Japanese large-footed bat (*Myotis macrodactylus*) [122] ddemonstrated that the wing membranes and bones of juvenile bats grow rapidly within the first few weeks after birth. Both wingspan and wing area (as shown in Figure 3) exhibit linear growth over 22 days until the bats initiate their first flight. During the first 20 days post-birth, the juveniles experience a significant increase in body mass and forearm length, after which the growth rate slows. However, parameters such as wingspan and wing area continue to increase until the first flight, enabling *Myotis emarginatus* to achieve sustained flight between 25 and 30 days, enhancing flight agility [162]. These developmental changes alter wing loading and morphology, improving flight efficiency and stability [150]. Further studies on *Hipposideros pomona* [121] corroborate these findings, emphasizing the importance of wing membrane and bone development during the early postnatal period, which significantly influences flight performance and maneuverability, allowing juveniles to achieve sustained flight within 25 to 30 days post-birth.

Due to the asynchrony between wing development and body growth, researchers commonly use parameters such as wing loading, wing area, and aspect ratio to quantify the progression and characteristics of flight development. J.A. McLean and J.R. Speakman investigated the changes in wingspan and wing area in brown long-eared bats (*Plecotus auritus*) [163], showing that wing morphology—specifically forearm length, wingspan, and wing area—grows more rapidly than body mass postnatally. Consequently, wing loading decreases sharply, reaching its lowest point around 30 days, during which juvenile bats exhibit more efficient, agile flight compared to adults. At this stage, flight speed, minimum energy demand speed, and maximum range speed are at their lowest, indicating energy-efficient and flexible flight. After 30 days, wing loading gradually increases, stabilizing near adult levels around day 49. Similar findings were reported by Elangovan et al. in a larger, non-echolocating species, the short-nosed fruit bat (*C. sphinx*) [120]. Their study revealed that the growth of the wings outpaced body mass gain within the first 35 days, with linear declines in wing loading. Wingspan and wing area stabilize after 45 days, during which the bats exhibit clumsy flight. Figure 4 synthesizes the relationship between postnatal growth (reflected by wing loading and aspect ratio) and flight performance across bat species from three representative families, highlighting both convergent trends and taxon-specific differences. This synthesis underscores that postnatal wing growth is a universal driver of bat flight performance, but the pace and duration of this process may be shaped by ecological niche.

## 4. Placing Wing and Powered Flight Development Within the Context of Echolocation Maturation

The ecological adaptability dimension of bat flyability is closely associated with the coordinated development of flight and echolocation. The mutual reinforcement of these two traits enables bats to efficiently adapt to nocturnal foraging and navigation environments. Undoubtedly, the ability to fly and use echolocation are two key strategies underpinning bats’ successful adaptation to diverse ecological environments [164,165]. Flight and echolocation are tightly intertwined throughout bat evolution, and although the academic debate regarding which of these abilities emerged first remains unresolved [166], it is clear that the development of flight and echolocation during the postnatal period is mutually reinforcing [167]. This relationship is particularly evident in laryngeal echolocating bats, where the maturation of both capabilities supports and enhances each other. Investigating the coordinated development of echolocation calls and flight abilities at different developmental stages is therefore crucial for understanding the behavioral ecology and physiological adaptability of bats.

The postnatal development of bats encompasses two core processes: the acquisition of flight capabilities and the maturation of echolocation calls [168]. Although different bat species vary in the specifics and timing of their postnatal development, studies consistently show a clear pattern of coordinated development between flight and echolocation during the first few weeks of life. For example, research on the Japanese large-footed bat (*M. macrodactylus*) [122] revealed that the rapid growth of wing membranes and bones during the early postnatal period coincides with the gradual refinement of echolocation calls, progressing from simple sound structures to complex, multi-band signals synchronized with flight performance. Similarly, in the big brown bat (*Eptesicus fuscus*) [169], the development of flight ability is accompanied by increasing complexity in echolocation calls. As the wing muscles and skeletal structures mature to support stable flight, echolocation signals evolve from simple, monotonic pulses to more sophisticated signals with directional properties and frequency modulation. Studies on the little brown bat (*Myotis lucifugus*) [170] further show that once juveniles achieve basic flight abilities, their echolocation calls undergo significant changes, particularly in terms of increased pulse frequency modulation and signal intensity.

Strengthening of the laryngeal structures, including the rapid firing of high-frequency calls, becomes critical for advanced echolocation. Carter and Adams’ research on the Jamaican fruit bat (*A*. *jamaicensis*) [171,172] highlights the synchronous development of the larynx—specifically, the calcification of the cricoid cartilage and the growth of the laryngeal muscles, particularly the cricothyroid muscle—with flight capabilities. During the “flap stage”, when bats begin to exhibit short-range powered flight, calcification initiates at the posterior region of the cricoid cartilage, gradually extending anteriorly. In the early “flight stage”, the entire cricoid cartilage becomes fully calcified, and this calcification persists through the “adult stage”. Comparative studies of the calcification levels in adult mice, wandering shrews (*Sorex vagrans*), and Jamaican fruit bats indicate that calcification is an adaptation supporting sonar emission. Although the laryngeal structures of wandering shrews also exhibit calcification, the lower degree of calcification suggests a lesser reliance on sonar compared to bats.

Predator–prey interactions further reinforce the coordinated development of flight and sensory traits, a key ecological dimension of bat flyability. Insectivorous bats (e.g., *E. fuscus* [169]) face predation pressure from aerial predators such as owls and falcons; in response, these bats have evolved a tight link between flight maneuverability and echolocation call plasticity; when detecting predator cues, they increase echolocation call frequency (enhancing obstacle detection) and switch to shorter, more frequent wingbeats (improving evasive flight). This integration of sensory and motor traits is not innate but develops postnatally. Juvenile *M. lucifugus* [170] gradually refine this predator avoidance flight–echolocation coordination during their first month of flight, with failure to do so increasing predation risk [170]. For fruit-eating bats [34], which face less aerial predation, although the link between flight and sensory traits is weaker, their flight adaptation prioritizes load-bearing capacity (for carrying fruit) over maneuverability, reflecting how predator–prey interactions shape the direction of flight–sensory integration.

Studies on the brown long-eared bat (*P. auritus*) [169] and the Mexican free-tailed bat (*T. brasiliensis*) [173] further confirm the coordinated development of flight behavior and echolocation. In both species, individuals receiving flight training within the first two weeks post-birth show accelerated maturation of their echolocation calls, suggesting that flight behavior may serve as a driving force for the learning process of echolocation. The frequency and intensity of flight training are directly correlated with the degree of echolocation call refinement, providing additional evidence for the mutually reinforcing relationship between flight and echolocation. During the early stages of echolocation development, the coordination and stability of flight significantly influence the effectiveness of echolocation. This is particularly evident in juvenile bats as they learn to fly, adjusting the frequency and intensity of their echolocation calls at varying altitudes and speeds to more efficiently detect prey and obstacles. This process not only helps bats optimize their flight strategies but also enhances the precision of their echolocation. Given these findings on the synergistic development of flight and echolocation, increasing research efforts are focusing on the maturation of flight capabilities within the context of echolocation call development.

## 5. Conclusions

Bats, as the only mammals capable of powered flight, exhibit remarkable adaptations in wing structure, muscle dynamics, and energy utilization, underpinned by complex interactions across molecular pathways, organs, and systems during development from embryogenesis to postnatal growth. This review synthesized knowledge on the developmental and evolutionary mechanisms of bat flight, emphasizing the integration of genetic, physiological, and biomechanical factors critical to understanding these processes. The interplay between flight and echolocation was explored, highlighting their co-evolution as a key adaptive strategy that enhances bats’ ecological versatility and species diversity.

## Figures and Tables

**Figure 1 biology-14-01524-f001:**
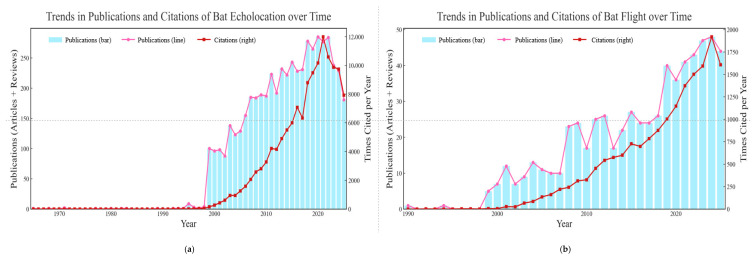
The two statistical charts illustrate the publication and citation trends for research on bat echolocation (**a**) and bat flight (**b**). A keyword search on the Web of Science database using “bat echolocation” (**a**) and “bat flight” (**b**) demonstrates that, although these terms do not capture the entirety of research on bat echolocation and flight capabilities, the existing literature is considerably more extensive for echolocation, whereas studies focused on bat flight remain comparatively limited. Search date: 1 September 2025.

**Figure 2 biology-14-01524-f002:**
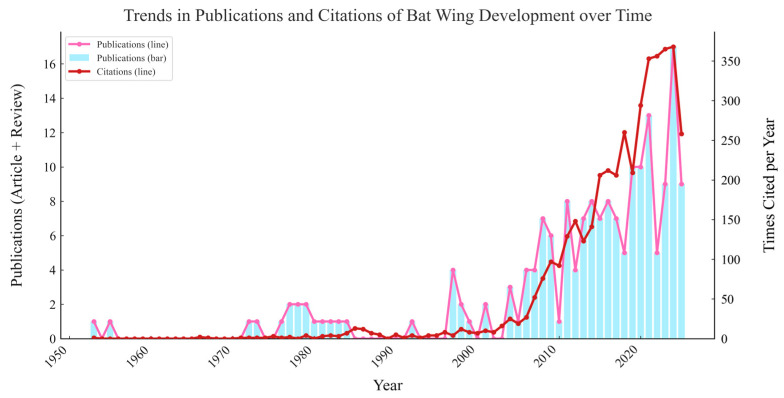
This figure summarizes the publication and citation trends for studies of bat wing development. Based on a keyword search in the Web of Science database. Search date: 1 September 2025.

**Figure 3 biology-14-01524-f003:**
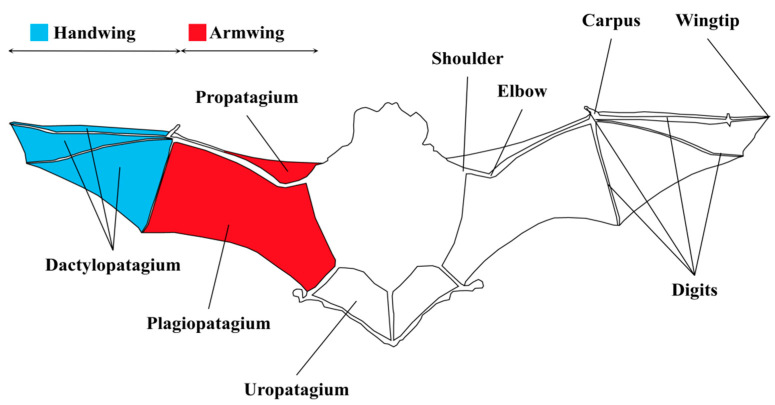
The schematic diagram of the bat wing structure. The wing membrane comprises several distinct components: the plagiopatagium, which extends between the body’s flank and the fifth digit; the propatagium, located between the shoulder and wrist; the dactylopatagium, supported by the fifth digit and the wingtip; and the uropatagium, which stretches between the legs and tail [44,47,62,63]. The bat wing is divided into two parts: the handwing and the armwing. The armwing primarily consists of the radius and ulna bones, covered by a flexible membrane, and adjusts its camber to different flight speeds to maintain stability and lift. In contrast, the handwing is formed by elongated finger bones and their overlying membrane, maintaining a relatively stable camber and playing a crucial role in generating thrust and additional lift [64].

**Figure 4 biology-14-01524-f004:**
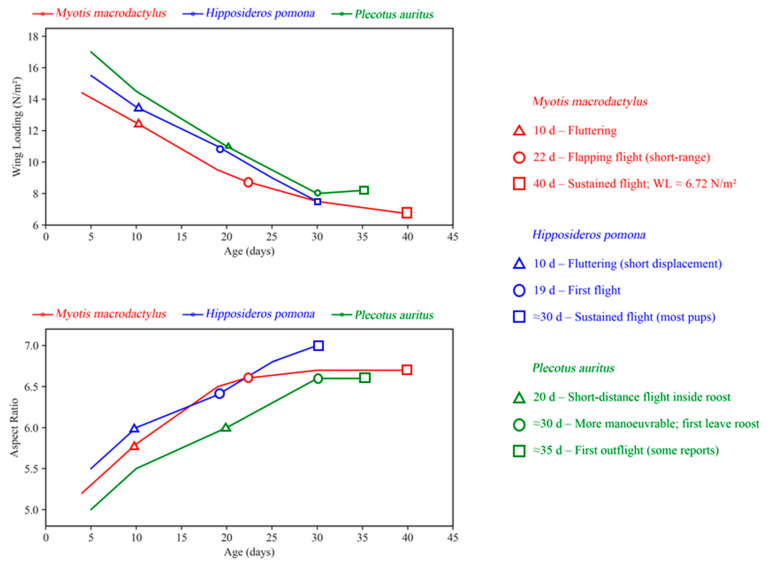
The postnatal changes in wing loading, aspect ratio and flight performance across three bat species.

**Table 1 biology-14-01524-t001:** Summary of studies retrieved to date using the keyword “bat wing development” in the WOS database search interface. This table primarily compiles three accessible studies, summarizing their research subjects, objectives, methods, and conclusions. Although this search term does not encompass all studies related to bat wing growth and development, it nonetheless indicates that research in this area remains relatively limited.

Family	Species	Examples	Objectives	Method	Conclusion
Phyllostomidae	*Carollia perspicillata*	[32]	Molecular basis of bat wing morphogenesis	Comparative gene expression and functional enhancer replacement experiments	Gene expression shifts drive wing specialization
Miniopteridae	*Miniopterus natalensis*	[33]	Wing bone mineralization patterns for flight adaptation	Biomechanical and proteomic profiling	Reduced mineralization enhances wing flexibility
Pteropodidae	*Pteropus poliocephalus*	[34]	Molecular profiling of bat wing development	RNA-seq and ChIP-seq of developing limbs	Gene regulation drives forelimb specialization
	*Pteropus hypomelanus*				

## Data Availability

Data are contained within the article or Appendix A.

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
