# Peer review of "What Factors Shape the Flyability in Bats?—The Perspective from Bat’s Wing Development"

_biology, 2025, doi:10.3390/biology14111524_

Round 1
Reviewer 1 Report
Comments and Suggestions for Authors
This manuscript by Zhang et al. provides a comprehensive review of factors shaping powered flight in bats, with an emphasis on wing development. It synthesizes embryonic, postnatal, structural, muscular, and metabolic aspects, while also discussing the evolution of flight and echolocation. The work is timely and relevant, especially given the relative underrepresentation of bat wing development studies compared to echolocation research. Figures and the supplementary table provide useful reference material, and the paper highlights important knowledge gaps for future research.
Overall, the manuscript is of interest and relevance to the readership of Biology. However, revisions are needed to clarify the conceptual framework, strengthen critical synthesis, and better integrate the supplementary materials.
Specific comments:
- The review identifies a real gap (wing development vs. echolocation research). However, the central concept of “flyability” is not clearly defined. Does it refer to aerodynamic efficiency, flight performance, flight capacity, developmental potential, or ecological adaptability? A sharper definition will help frame the paper more coherently.
- The introduction is strong and nicely positions the gap between echolocation and wing development research. Figures 1 & 2 are useful but could be interpreted more critically. What do publication trends reveal about research biases? Please interpret why wing development has historically been neglected.
- Table S1 is valuable but currently stands apart from the main text. Grouping the studies into thematic categories (such as aerodynamics, energetics & metabolism, neuro-sensory integration) and synthesizing trends would strengthen its contribution.
- Molecular and developmental aspects are covered in detail (e.g., MEIS2, TBX3, Wnt5a pathways). However, ecological and biomechanical dimensions (such as habitat constraints, predator–prey interactions, climate effects) receive less emphasis. Balancing these sections would enhance the review’s scope.
- Much of the text summarizes existing studies but does not sufficiently evaluate the findings, taking into account conflicting evidence or methodological limitations. For example, fossil interpretations of Onychonycteris finneyi (flight first vs. echolocation first) are presented descriptively but not critically discussed. The authors could expand on the ongoing debate, such findings from Veselka et al. 2010, where they maintain finneyi may have been capable of echolocation, and follow up with Simmons et al., 2010 who argue against this. Additionally, acknowledging that soft-tissue structures, such as those critical for echolocation, are rarely preserved in fossils would underline the uncertainty and enrich the evolutionary discussion.
- In a similar example to point #5 above, in Section 2.3, energy metabolism in bats is presented as largely fat-driven, but dietary guild differences could be contrasted. The authors miss an opportunity to highlight ecological diversity in bat energetics. This section could be expanded to distinguish between insectivorous bats, which rely primarily on fat metabolism, and nectarivorous/frugivorous bats, which are capable of fueling flight directly with exogenous sugars. Contrasting these strategies would strengthen the argument that bat “flyability” is shaped not just by wing morphology, but also by diet and ecological niche.
- In the energy metabolism section, consider adding more explicit comparison with birds, since both bats rely on fat metabolism during sustained flight. This would broaden appeal and highlight convergent evolution.
- Postnatal development section (section 3.2) is very strong. Consider adding a synthesis model of growth vs. flight performance across species (Myotis macrodactylus, Hipposideros pomona, Plecotus auritus).
- The section on flight–echolocation interplay (section 4) is great. It would be useful to add figure showing parallel timelines of wing and echolocation development.
- Line 167 has an incomplete sentence: “The lack of robust transitional fossil records [41].”
Author Response
This manuscript by Zhang et al. provides a comprehensive review of factors shaping powered flight in bats, with an emphasis on wing development. It synthesizes embryonic, postnatal, structural, muscular, and metabolic aspects, while also discussing the evolution of flight and echolocation. The work is timely and relevant, especially given the relative underrepresentation of bat wing development studies compared to echolocation research. Figures and the supplementary table provide useful reference material, and the paper highlights important knowledge gaps for future research. Overall, the manuscript is of interest and relevance to the readership of Biology. However, revisions are needed to clarify the conceptual framework, strengthen critical synthesis, and better integrate the supplementary materials. Response: We greatly appreciate your professional review of our manuscript. Your comments and suggestions are highly professional and valuable, providing significant support for improving the quality of our manuscript. We have carefully and thoroughly revised the manuscript in response to each of your comments and suggestions. The following context will indicate detailed changes we have made in the revised manuscript. Thank you again for your valuable feedback and the time you have invested in reviewing our manuscript. Specific comments: 1. The review identifies a real gap (wing development vs. echolocation research). However, the central concept of “flyability” is not clearly defined. Does it refer to aerodynamic efficiency, flight performance, flight capacity, developmental potential, or ecological adaptability? A sharper definition will help frame the paper more coherently. Response 1: We highly appreciate this suggestion and have revised the manuscript accordingly. Firstly, we have added a clear definition of "flyability" in the Simple Summary, please refer to lines 14-15. Secondly, to integrate "flyability" into the framework of key sections and reinforce logical continuity, we have added relevant descriptions at the beginning of Section 2, Section 3, and Section 4, please refer to lines 225-228, lines 544-548 and lines 696-699 2. The introduction is strong and nicely positions the gap between echolocation and wing development research. Figures 1 & 2 are useful but could be interpreted more critically. What do publication trends reveal about research biases? Please interpret why wing development has historically been neglected. Response 2: Thank you very much for this suggestion. We have supplemented the Introduction section with related explanations to strengthen the coherence and depth of the Introduction. Please refer to lines 82-93 and lines 94-103. We hope these revisions could enhance the critical analysis of Figures 1 & 2, clarify the research biases in bat biology, and explain the historical neglect of wing development research. It should be noted that we have identified that Figure 1 submitted in the initial manuscript was not the final version, and it has been replaced with the correct and updated version. 3. Table S1 is valuable but currently stands apart from the main text. Grouping the studies into thematic categories (such as aerodynamics, energetics & metabolism, neuro-sensory integration) and synthesizing trends would strengthen its contribution. Response 3: Thank you for this suggestion. Accordingly, we have revised Table S1 and its associated discussion. Specifically, we have reorganized the studies previously listed in Table S1 into five clear thematic categories aligned with the core focus of our review, including Aerodynamics and kinematics, Energetics and metabolism, Morphology and development, Ecology and Neuro-sensory Integration. For further, we added a concise synthesizing paragraph in the main text (immediately following the first mention of Table S1) to summarize key trends. Please refer to lines 112-128. 4. Molecular and developmental aspects are covered in detail (e.g., MEIS2, TBX3, Wnt5a pathways). However, ecological and biomechanical dimensions (such as habitat constraints, predator–prey interactions, climate effects) receive less emphasis. Balancing these sections would enhance the review’s scope. Response 4: Thank you very much for this valuable suggestion. We have revised the manuscript with targeted content in three key sections to address the ecological and biomechanical dimensions you highlighted. In details, in Section 2.3, we added content linking energy metabolism to ecological and climate effects; in Section 2.4, we added analyses of biomechanical-ecological links; in Section 4, we integrated content on predator–prey interactions and their role in flight-sensory integration. Please refer to lines 430-456 and lines 529-541 and lines 738-750. We hope these revisions could ensure the review now balances molecular/developmental, ecological, and biomechanical perspectives, providing a more comprehensive analysis of the factors shaping bat flyability. 5. Much of the text summarizes existing studies but does not sufficiently evaluate the findings, taking into account conflicting evidence or methodological limitations. For example, fossil interpretations of Onychonycteris finneyi (flight first vs. echolocation first) are presented descriptively but not critically discussed. The authors could expand on the ongoing debate, such findings from Veselka et al. 2010, where they maintain finneyi may have been capable of echolocation, and follow up with Simmons et al., 2010 who argue against this. Additionally, acknowledging that soft-tissue structures, such as those critical for echolocation, are rarely preserved in fossils would underline the uncertainty and enrich the evolutionary discussion. Response 5: Thank you very much for this suggestion. Accordingly, we have revised the relevant section (Section 1, Introduction) to enhance the critical discussion of Onychonycteris finneyi fossil interpretations and highlight key challenges in inferring soft-tissue-dependent traits from fossils. Please refer to lines190-200. We have also updated three additional key sections of the manuscript to enhance critical analysis across different dimensions of bat flight research, ensuring consistency with your feedback to move beyond descriptive summarization. Please refer to lines 456-462 and lines 526-528 and lines 620-621. 6. In a similar example to point #5 above, in Section 2.3, energy metabolism in bats is presented as largely fat-driven, but dietary guild differences could be contrasted. The authors miss an opportunity to highlight ecological diversity in bat energetics. This section could be expanded to distinguish between insectivorous bats, which rely primarily on fat metabolism, and nectarivorous/frugivorous bats, which are capable of fueling flight directly with exogenous sugars. Contrasting these strategies would strengthen the argument that bat “flyability” is shaped not just by wing morphology, but also by diet and ecological niche. Response 6: Thank you for this suggestion. We fully agree with this suggestion and have revised Section 2.3 to explicitly distinguish energy strategies between insectivorous bats and nectarivorous/frugivorous bats, strengthening the argument that bat “flyability” is shaped by diet and ecological niche alongside wing morphology. Please refer to lines 430-462 in the revised manuscript. 7. In the energy metabolism section, consider adding more explicit comparison with birds, since both bats rely on fat metabolism during sustained flight. This would broaden appeal and highlight convergent evolution. Response 7: Thank you for this suggestion. Accordingly, we have integrated this comparison into the energy metabolism section, please refer to lines 404-406. And we chose to frame this comparison concisely to align with the core focus of our review, while still acknowledging the evolutionary convergence you highlighted. Thank you again for this suggestion. 8. Postnatal development section (section 3.2) is very strong. Consider adding a synthesis model of growth vs. flight performance across species (Myotis macrodactylus, Hipposideros pomona, Plecotus auritus). Response 8: Thank you for this suggestion. According to your suggestion, we have added Figure 4 and related description to Section 3.2, which synthesizes the model of growth (mainly reflected by wing loading and aspect ratio) vs. flight performance across bat species from three representative families. This figure specifically includes species relevant to our manuscript. Please refer to figure 4 and lines 659-692 in this revised manuscript. 9. The section on flight–echolocation interplay (section 4) is great. It would be useful to add figure showing parallel timelines of wing and echolocation development. Response 9: Thank you sincerely for this suggestion. We fully appreciate the value of this visualization in highlighting the coordinated maturation of these two key traits, and we have carefully explored feasible approaches to implement your recommendation. However, after thorough evaluation of the biological complexity and cross-species variability inherent in our review’s scope, we encountered challenges that prevent us from creating a meaningful, non-misleading figure. Details we wish to share with you. First, divergent developmental trajectories across bat species make cross-species timeline integration unworkable. As outlined in our manuscript, existing studies confirm the coordinated development of echolocation and flight in bats, but the maturation timeline and trajectory vary significantly among species. Plotting these disparate timelines on a single figure would either oversimplify critical species-specific differences (e.g., compressing timelines to fit a uniform scale) or create visual clutter that obscures the "coordinated development" theme you aim to highlight. Second, the multi-dimensional nature of echolocation and flight makes quantitative visualization infeasible. Echolocation maturation involves non-interchangeable parameters—including call frequency (kHz), pulse duration (ms), and signal intensity (dB)—each following independent developmental paths. Reducing these to a single "echolocation strength" metric for a timeline would erase biological nuance. Similarly, flight ability encompasses wing loading (N/m²), maneuverability (turn radius), and flight stability, many traits that often mature asynchronously. A line chart or scatter plot would force arbitrary prioritization of one parameter over others, leading to an incomplete or misleading representation. While we cannot implement the figure for the current review, we strongly endorse the merit of your suggestion. We fully agree it is an exceptionally effective way to present findings from single-species focused studies, where trait parameters and developmental timelines are consistent, allowing for clear, precise visualization. In fact, your input has directly informed our ongoing research: We are currently investigating the coordinated development of flight and echolocation in Vespertilio sinensis, having completed echolocation sound recording and corresponding flight performance experiments. In the subsequent data analysis and result presentation for this single-species study, we plan to incorporate the parallel timeline visualization you suggested, fully leveraging its strengths to showcase trait synergy. We sincerely thank you for this insightful suggestion, which has not only guided our reflection on the current manuscript but also enhanced the design of our future work. We hope this explanation addresses your feedback effectively. 10. Line 167 has an incomplete sentence: “The lack of robust transitional fossil records [41].” Response 10: We have completed this sentence in the revised manuscript, please refer to lines 211-213, and we have reviewed the entire manuscript to avoid similar missing. Thank you for this suggestion.

Reviewer 2 Report
Comments and Suggestions for Authors
Bats (Chiroptera) are a unique group of mammals possessing the ability to perform active flight –a functional characteristic that distinguishes them from all other members of their class. The evolution of flight in bats has been accompanied by profound morphological, physiological, and molecular changes affecting limb structure, muscle dynamics, and energy metabolism. Studying these adaptations is of significant interest for understanding the mechanisms underlying the evolution of complex forms of locomotion, as well as for uncovering the relationships between molecular pathways, organ systems, and developmental processes from embryogenesis to the postnatal period. The article "What Factors Shape the Flyability of Bats? - The Perspective 2 from Bat's Wing Development" presents a detailed review of the mechanisms underlying the development and evolution of bat flight, with an emphasis on the integration of genetic, physiological, and biomechanical factors.
In this review, the authors attempt to cover a fairly broad range of topics, including the developmental and evolutionary mechanisms underlying patagium formation in gliding or flying mammals, the effectiveness of energy conservation and antioxidant defense, the specialization of flight muscles and wings for active flight, their development during the embryonic and postembryonic periods, and the definition of wing development and active flight in the context of echolocation development. The results presented in the article may be of great interest – both from the standpoint of fundamental science and for related applied fields. This work can serve as a basis for further comparative evolutionary, genetic, and physiological studies aimed at understanding the principles of complex adaptations and the convergent evolution of flight in mammals. Therefore, I have no doubt about the relevance of this work.
The manuscript submitted for review is clear, relevant to the subject matter, and well structured. The authors were able to present complex and disparate material in a high-quality manner in an accessible form. The statements and conclusions are consistent and supported by citations. The sources cited are relevant to the issues under consideration, most of them are recent publications, and they are adequate to the content. The scope of the sources reviewed is sufficient to cover the issues raised in the most comprehensive manner. The figures and tables presented are clear and easy to interpret. However, a few minor comments arose during the reading.
- Lines 67-70. The authors provide a classification of signals: "Based on differences in vocalization methods and frequencies, echolocating bats can be categorized into frequency modulation (FM) bats, constant frequency-frequency modulation (CF-FM) bats, click bats (Click), and click-like bats (Click-like) [19,20]." The latter are referred to as click bats (Click). However, there is a more established designation for these types of signals. They should be called frequency-modulated, quasi-constant frequency (FM-QCF) pulses.
- Lines 152-155 and 162-165. Clearly, the body structure of Onychonycteris is already adapted for flight, and it turns out to be the most primitive bat known to date. However, it must be acknowledged that Onychonycteris certainly did not have developed echolocation; the proportions of its limb bones are more reminiscent of those of climbing animals than bats. Powerful flight is out of the question here, as the authors point out. The bones of its metacarpus and fingers, although noticeably elongated, are still less so than those of Icaronycteris. Onychonycteris's flight apparently served a transport function, flying from tree to tree and was not distinguished by either speed or maneuverability. I think the authors should pay attention to this.
- Lines 143-144. The cited source gives the age of the find as 52.5 million years.
- Lines 278-284, 429-432. In my opinion, the attachment site of the proximal part of the potagium to the forelimb should also be noted. In gliders, the potagium attaches to the carpus, allowing full use of the hand for support and movement along the substrate. In bats, unlike in gliders, it forms the propelling part of the wing, stretched between digits 2-5 – the dactylopatagium. It would be logical to discuss in more detail the role of this part of the wing in flight and its modifications in different Chiroptera species.
Thus, the work under discussion can be characterized as follows:
Research topic: relevant.
The review may be of interest to a wide range of specialists.
Overall conclusion: recommended for publication after the authors make revisions based on the reviewer's comments.

Author Response
Bats (Chiroptera) are a unique group of mammals possessing the ability to perform active flight –a functional characteristic that distinguishes them from all other members of their class. The evolution of flight in bats has been accompanied by profound morphological, physiological, and molecular changes affecting limb structure, muscle dynamics, and energy metabolism. Studying these adaptations is of significant interest for understanding the mechanisms underlying the evolution of complex forms of locomotion, as well as for uncovering the relationships between molecular pathways, organ systems, and developmental processes from embryogenesis to the postnatal period. The article "What Factors Shape the Flyability of Bats? - The Perspective 2 from Bat's Wing Development" presents a detailed review of the mechanisms underlying the development and evolution of bat flight, with an emphasis on the integration of genetic, physiological, and biomechanical factors.
In this review, the authors attempt to cover a fairly broad range of topics, including the developmental and evolutionary mechanisms underlying patagium formation in gliding or flying mammals, the effectiveness of energy conservation and antioxidant defense, the specialization of flight muscles and wings for active flight, their development during the embryonic and postembryonic periods, and the definition of wing development and active flight in the context of echolocation development. The results presented in the article may be of great interest – both from the standpoint of fundamental science and for related applied fields. This work can serve as a basis for further comparative evolutionary, genetic, and physiological studies aimed at understanding the principles of complex adaptations and the convergent evolution of flight in mammals. Therefore, I have no doubt about the relevance of this work.
The manuscript submitted for review is clear, relevant to the subject matter, and well structured. The authors were able to present complex and disparate material in a high-quality manner in an accessible form. The statements and conclusions are consistent and supported by citations. The sources cited are relevant to the issues under consideration, most of them are recent publications, and they are adequate to the content. The scope of the sources reviewed is sufficient to cover the issues raised in the most comprehensive manner. The figures and tables presented are clear and easy to interpret. However, a few minor comments arose during the reading.
Response: We are deeply grateful for your positive evaluation of our manuscript and your thoughtful feedback. Your recognition of the manuscript means a great deal to us. We fully agree with your suggestions and have implemented targeted revisions to further enhance the manuscript’s clarity and consistency. Thank you again for your valuable feedback and the time you have invested in reviewing our manuscript.
- Lines 67-70. The authors provide a classification of signals: "Based on differences in vocalization methods and frequencies, echolocating bats can be categorized into frequency modulation (FM) bats, constant frequency-frequency modulation (CF-FM) bats, click bats (Click), and click-like bats (Click-like) [19,20]." The latter are referred to as click bats (Click). However, there is a more established designation for these types of signals. They should be called frequency-modulated, quasi-constant frequency (FM-QCF) pulses.
Response 1: Thank you for this suggestion. Accordingly, we have revised the description using “frequency-modulated, quasi-constant frequency (FM-QCF)” instead of “Click-like”, to align with the widely accepted designation for echolocation signal types. Please refer to line 74.
- Lines 152-155 and 162-165. Clearly, the body structure of Onychonycteris is already adapted for flight, and it turns out to be the most primitive bat known to date. However, it must be acknowledged that Onychonycteris certainly did not have developed echolocation; the proportions of its limb bones are more reminiscent of those of climbing animals than bats. Powerful flight is out of the question here, as the authors point out. The bones of its metacarpus and fingers, although noticeably elongated, are still less so than those of Icaronycteris. Onychonycteris's flight apparently served a transport function, flying from tree to tree and was not distinguished by either speed or maneuverability. I think the authors should pay attention to this.
Response 2: Thank you for your valuable insight regarding the functional traits of Onychonycteris finneyi. We fully agree that clarifying its primitive flight characteristics and limb morphology is critical for accurately contextualizing early bat flight evolution. According to your suggestion, we have supplemented the discussion on Onychonycteris finneyi (lines 185-190) to explicitly address its limited flight capabilities and climbing-like limb proportions, while maintaining consistency with the fossil evidence cited in the manuscript.
- Lines 143-144. The cited source gives the age of the find as 52.5 million years.
Response 3: Done. Thank you very much for this suggestion.
- Lines 278-284, 429-432. In my opinion, the attachment site of the proximal part of the potagium to the forelimb should also be noted. In gliders, the potagium attaches to the carpus, allowing full use of the hand for support and movement along the substrate. In bats, unlike in gliders, it forms the propelling part of the wing, stretched between digits 2-5 – the dactylopatagium. It would be logical to discuss in more detail the role of this part of the wing in flight and its modifications in different Chiroptera species.
Response 4: Thank you very much for this suggestion. We have carefully incorporated your suggestion into the two specified sections of the manuscript. For lines 278-284 (in previous manuscript), we expanded the comparison between bats and gliding mammals to explicitly address the proximal patagium’s attachment site and function, a key structural distinction you emphasized, please refer to 326-339 in this revised manuscript. For Lines 429-432 (in previous manuscript), we built on the dactylopatagium’s introduction by discussing its modifications across Chiroptera and also to link this structure to ecological adaptation. Please refer to lines and lines 526-541 in the revised manuscript.

Reviewer 3 Report
Comments and Suggestions for Authors
The study is well-designed, focusing on the characteristics of powered flight in bats from the aspects of specialised wing structure, physiological adaptations for flight, and molecular regulation of wing development. Tables and graphs are in line with the investigated topic and give a good visualisation of the study problem. I have some minor comments:
- In lines 58-59: It's better to mention primitive echolocation in a few species, e.g. See https://royalsocietypublishing.org/doi/10.1098/rspb.2021.1714
- In Table 1 and S1, in the column marked as 'Species', delete the first letter of the genus because it is already written in the column 'Genus'.
< !--EndFragment -->
Author Response
The study is well-designed, focusing on the characteristics of powered flight in bats from the aspects of specialised wing structure, physiological adaptations for flight, and molecular regulation of wing development. Tables and graphs are in line with the investigated topic and give a good visualisation of the study problem. I have some minor comments:
Response: We are deeply grateful for your positive evaluation of our manuscript and your constructive feedback. We fully agree with your suggestions and have implemented targeted revisions to further improve the manuscript. Thank you again for your valuable feedback and the time you have invested in reviewing our manuscript.
In lines 58-59: It's better to mention primitive echolocation in a few species, e.g. See https://royalsocietypublishing.org/doi/10.1098/rspb.2021.1714
Response: Thank you for this suggestion. We have added the corresponding description and included a new reference. Please refer to lines 62-63 in the revised manuscript.
In Table 1 and S1, in the column marked as 'Species', delete the first letter of the genus because it is already written in the column 'Genus'.
Response: Thank you for this suggestion. In consideration that the "Species" column already contains the genus information, we have revised Tables 1 and Table S1 with reference to your suggestion, while adopting a slightly different approach: instead of modifying the "Species" column entries, we have deleted the entire "Genus" column. This revision streamlines the table layout while preserving all critical taxonomic details, ensuring readers can quickly associate each species with its corresponding genus without redundant content. We appreciate your suggestion, which guided us to optimize the table structure for better readability.

Round 2
Reviewer 1 Report
Comments and Suggestions for Authors
The revised manuscript by Zhang et al. shows substantial improvement. I thank them for their thoroughness. The authors have successfully clarified and defined the concept of “flyability,” which now provides a coherent framework throughout the paper. The reorganization of Table S1 into thematic categories strengthens the structure and accessibility of the review. The new Figure 4 effectively synthesizes postnatal wing growth and flight performance across species.
The discussion of Onychonycteris finneyi has been significantly enhanced, now including references to the ongoing debate about echolocation capability and highlighting soft-tissue preservation limitations. The section on energy metabolism has been notably expanded, contrasting fat-driven and sugar-driven strategies among bat dietary guilds and linking metabolism to ecology and climate. These additions greatly improve the balance between developmental, physiological, and ecological perspectives.
Overall, the manuscript is now comprehensive, critical, and well written.